# Construction and Validation of an Analytical Grid about Video Representations of Suicide (“MoVIES”)

**DOI:** 10.3390/ijerph16152780

**Published:** 2019-08-03

**Authors:** Christophe Gauld, Marielle Wathelet, François Medjkane, Nathalie Pauwels, Thierry Bougerol, Charles-Edouard Notredame

**Affiliations:** 1CHU Grenoble Alpes, F-38000 Grenoble, France; 2Department of Public Health, CHU Lille, F-59000 Lille, France; 3Fédération de Recherche en Psychiatrie et Santé Mentale des Hauts-de-France, F-59000 Lille, France; 4Department of Child and Adolescent Psychiatry, CHU Lille, F-59000 Lille, France; 5SCALab, CNRS, UMR 9193, F-59000 Lille, France

**Keywords:** suicide, Werther effect, psychology of cinema, identification, emotion, vulnerability, imitation

## Abstract

*Background.* Exposure to fictional suicide scenes raises concerns about the risk of suicide contagion. However, researchers and clinicians still lack empirical evidence to estimate this risk. Here, we propose a theory-grounded tool that measures properties related to aberrant identification and suicidal contagion of potentially harmful suicide scenes. *Methods.* The items of the *Movies and Video: Identification and Emotions in reaction to Suicide* (MoVIES) operationalize the World Health Organization’s recommendations for media coverage of suicide, and were adapted and completed with identification theory principles and cinematographic evidence. Inter-rater reliability (Cohen’s kappa) and internal consistency (Cronbach’s alpha) were estimated and optimized for two series of 19 and 30 randomly selected movies depicting a suicide scene. The validity of the scale in predicting identification with the suicidal character was tested in nine unknowledgeable participants who watched seven suicide movie scenes each. *Results*. The MoVIES indicated satisfying psychometric properties with kappas measured at 0.7 or more for every item and a global internal consistency of [α = 0.05]. The MoVIES score significantly predicted participants’ strength of identification independently from their baseline empathy ((β = 0.20), *p* < 0.05). *Conclusions*. The MoVIES is available to scholars as a valid, reliable, and useful tool to estimate the amount of at-risk components of fictional suicidal behavior depicted in films, series, or television shows.

## 1. Introduction

The interference of video productions in public health issues occasionally arises in public debate (for example, as reported in press articles, such as in the New Yorker in May 2019: “Netflix and Suicide: The Disturbing Example of 13 Reasons Why”) [1], prompting scholars to seize the matter in terms of research and prevention. Along with violence [2], addiction [3], and public health in general [4], one of the most paradigmatic, tangible, and perhaps disturbing impacts of cinema and TV shows on people concerns suicidal behaviors [5,6,7]. Although initially described [8] and repeatedly demonstrated in the press media [9,10,11], the Werther Effect (WE) (i.e., the systematic increase of suicide rates following incautious media coverage of suicide news stories [12]) has been extended to television [13,14,15]. Based on a systematic review of 11 ecological studies, Pirkis and Blood argued that the association between television broadcasts of suicide and rates of suicidal behaviors is consistent and strong [16]. Conversely, evidence has been emerging for almost a decade that some reports of suicidal events could be associated with a reduction of suicide rates via the so-called Papageno effect (PE) [17,18,19].

Although less unequivocal [15,16], evidence confirms that broadcasts of fictional suicides tangibly impact viewers. For instance, audience reception studies based on pseudo-experimental designs suggest that exposure to movies featuring suicide scenes can increase suicidality in already vulnerable individuals [20,21]. Recently, the release of the Netflix show *13 Reasons Why*, which depicts the suicidal process of an American adolescent girl, provided an ecological example of the effect of a video suicide narrative at the population level. Indeed, several authors cautioned about the serious—albeit potentially mixed [22,23,24]—impact of the series on youths’ mental health [25]. In the months following the broadcast, Niederkrothentaler et al. found that suicide rates among those aged 10–19 years were 13% higher than expected from secular trends [14]. Similarly, Bridge et al. [26] evidenced that the release of *13 Reasons Why* was associated with a significant increase in monthly suicide rates among youth in the US aged 10 to 17 years (incidence rate ratio (IRR), 1.29; 95% CI, 1.09–1.53). The authors warned about the exposure of children and adolescents to the series.

Among the hypotheses proposed to account for suicidal contagion at the individual level, identification may be one integrative and heuristically effective construct [27,28]. Identification relates to a psychosocial process in the interplay between neuroscience, cognitive, and behavioral theories, and developmental conceptualizations. From a phenomenological viewpoint, identification qualifies the experience of temporarily borrowing the perspective of another person (whether real or fictive), while weakening (although not totally “dissolving”) self-awareness [29]. From this perspective, identification finds a relevant translation in the neuroscience realm under the notion of empathy [30]. According to recent understandings, empathy results from the dynamic integration of two opposite processes: (1) Embodied simulation of others’ emotions and representations leading to first-/third-person confusion; and (2) mentalization, which enables discriminating one’s own state of mind from that inferred for others [31,32].

The simulation component is heavily involved in the affective arousal triggered by emotionally charged video scenes, while mentalization contributes to tempering emotional contagion [33]. Interestingly, identification may translate into the behavioral repertoire through social learning. According to Bandura’s theory [34], the probability of one individual adopting a behavior observed in a character is correlated with the strength of identification with this character. In suicidal contagion, embodied simulation may be insufficiently balanced by mentalization, leading to such confusion with the character that it leads to an excessive emotional reaction and maladaptive imitation. Conversely, the observation of coping, help-seeking, or help-offering behaviors may lead to protective effects through a similar process.

The strong debates emerging in the scientific community after the release of *13 Reasons Why* highlight the need for better insights into the impact of video suicide in terms of contagion. Although recent studies from Niederkronthentaler et al. [14] and Bridge et al. [26] provide new epidemiological evidence for the causal influence of fictional suicide on suicide rates, the mechanisms that underlie this macro-individual observation remain poorly known. Contrary to the press media for which screening tools exist [35,36], both ecological and audience reception studies on video fictional suicide are hindered by the absence of valid instruments to analyze productions.

In this paper, we describe the development and validation process of the *Movies and Video: Identification and Emotions in Reaction to Suicide* (MoVIES). The MoVIES is a theory-oriented analytic tool designed to quantitatively characterize video scenes depicting a suicidal gesture regarding mechanisms involved in the suicidal contagion process, with a special focus on identification.

## 2. Materials and Methods

An overview of the construction and validation process is provided in Figure 1. Below, we describe in detail the successive steps.

### 2.1. Construction of the MoVIES

The final aim of the MoVIES is to provide a quantitative value summarizing the at-risk properties of video productions depicting suicidal behaviors in terms of suicidal contagion. This construct is purposely holistic and theory-oriented, as it assumes that both the Werther and Papageno effects are at least partially mediated by the identification process.

In accordance with this comprehensive approach, we extracted items for the MoVIES from two complementary sources. The first set of items formally qualifies the video sequence to estimate the strength of the embodied simulation it is expected to induce. To compose this set, we reviewed film studies on cinema related to the phenomena of identification, embodied simulation, and the character–spectator relationship [37,38,39]. We extracted the cinematographic and soundtrack properties for which the association with the viewer’s immersive impression received greater evidence. For instance, Braudy [40] suggested that the realism of a character or scene eases identification, while Bordwell [41] showed that close-up shots provide an impression of physical proximity and intimacy with the character. The second set of items was intended to capture properties associated with the probability that the viewer would adopt imitative behaviors in response to the content of the scene. This set consists of a translation and operationalization of the World Health Organization’s (WHO) recommendations for responsible media coverage of suicide (WHO, 2017) [42], as applied to the video media. For instance, the MoVIES probes whether the film displays help resources or whether suicidal behavior is glamorized. Importantly, the content items consider that the films can non-exclusively lead to risk-taking or protective behavior (i.e., induce the Werther and/or Papageno effects) as a result of the identification process.

Each item was rated as 1 (when the assertion is true) or 0 (when the assertion is false). The total score of the grid is obtained by summing the scores of individual items. To preserve the coherence of the construct, we reversed the rating for items referring to the Papageno effect, so that a higher score on the MoVIES indicates a greater amount of at-risk components in the video.

### 2.2. Sample of Suicide Scenes

The sample of suicide scenes used to test reliability was extracted from two databases. (1) The Internet Movie Database (IMDb) is the largest free collaborative film database in the world (2015). The platform inventories nearly 5.3 million films that users (83 million are registered) can index according to their topic, style, or type. Following previously used methodology [43], we applied the keyword “suicide” to retrieve films tagged as dealing with suicidal behaviors. (2) The book *Suicide Movies: Social Patterns 1900–2009* [44] catalogs 1158 films from 1900 to 2009, depicting a scene of suicide. To date, this book offers the most comprehensive census of suicide-related films.

One at a time, 30 films were randomly extracted from the merged databases and screened for eligibility. Films were selected based on whether they depicted at least one scene in which a character dies by or attempts suicide, have been broadcast to the public (irrespective of the channel), and if the original version was in English. Excluded references were duplicates, documentaries, and movies not explicitly depicting suicidal behavior. The consecutive screening procedure was repeated until the optimization and validation sample was constituted.

### 2.3. Optimization of the MoVIES

#### 2.3.1. Linguistic Optimization

For comprehensibility, the first version of the MoVIES constructed by the authors was submitted to a focus group comprising 12 graduated medical students, naïve regarding the topic and purpose of the study. The students discussed the suitability, clarity, and precision of each item. Discussions were supervised by the main investigator (CG) until reaching a consensus about whether and which semantic and syntactic corrections were necessary. No items were added or removed. This refining procedure resulted in a test version of the grid comprising 41 items.

Furthermore, written consent was obtained. No national regulatory ethics authorization was necessary, because this study did not involve clinical subjects or any medical device. However, we obtained official approval from the local ethics committee of the Grenoble Hospital Center.

#### 2.3.2. Inter-Rater Reliability Optimization

As a first step, the MoVIES was tested on a limited sample of suicide scenes (*N* = 19). The two raters were placed in similar viewing conditions. They filled the grid blind to each other immediately after watching each film excerpt. After this first series, we estimated the inter-rater reliability of each item by measuring the corresponding Cohen’s kappas. The agreement of items with a kappa lower than 0.6 was considered insufficient [45]. Based on their rating experience, the investigators discussed further operationalization and clarification of problematic items. If no consensus was reached, a third investigator arbitrated the corrections to be made.

### 2.4. Statistical Analysis

#### 2.4.1. Reliability and Internal Consistency

We estimated the reliability of the MoVIES in terms of inter-judge fidelity and internal consistency. To do so, two investigators concurrently used the grid on 30 random samples of suicide scenes.

*Internal consistency*: The internal consistency of the MoVIES was evaluated by estimating Cronbach’s alpha coefficient of the sample of validation. A Cronbach’s alpha greater than 0.70 was considered “satisfactory” [46]

*Inter-expert reliability:* Inter-expert reliability was evaluated in two successive series of movies by calculating Cohen’s kappa for each item.

#### 2.4.2. Predictive Validity

We aimed to test the ability of the MoVIES score to predict induced identification in viewers independently from emotional arousal and baseline empathy. A multiple linear regression was conducted, predicting the identification (Cohen Identification Questionnaire) for the score of the MoVIES and adjusting for the empathy scale (Empathy Questionnaire), emotional valence, and the emotional arousal scale (Self-Assessment Manikin). Data were available for nine students: Four women and five men (authors were excluded). They watched seven suicide scenes each at a rate of one to two scenes per week. The scenes, which lasted two to six minutes, were randomly selected from the reliability-testing sample. The main investigator (CG) explained the synopsis of the films before playing the corresponding scenes and supervised the viewing sessions, which lasted 45 minutes. Participants completed the Empathy Questionnaire prior to the viewing session, and filled in Cohen’s identification grid and the Self-Assessment Manikin immediately after each suicide scene.

The following instruments were used in the model:*Cohen Identification Questionnaire (CIQ)* [47,48]: The CIQ is a self-administered, 10-item questionnaire that objectifies how much a viewer identifies with a film character. For each question, participants are invited to rate their agreement with the corresponding statement on a five-point Likert scale. The total score ranges from 0 to 40. In the present study, we used an ad hoc French version of the scale. To our knowledge, this is the only scale to measure identification. It has been used in several audience reception studies [20,27];*Empathy Questionnaire* (EQ) [49]: With α = 0.92, the EQ has been suggested as one of the most robust and consistent scales to measure empathetic abilities. It consists of 60 questions, to which the participant responds on a Likert scale ranging from “Strongly disagree” to “Strongly agree”. A validated French version is available in Delpechitre et al. [50] and Sonié and Robinson [51];*Self-Assessment Manikin* (SAM) [52]: The SAM is a self-administered scale that non-verbally measures (i.e., with pictograms) a person’s emotional valence and arousal in response to a stimulus [53]. The figures in the first row range from a frown to a smile, representing the valence dimension. The second row depicts figures ranging from a peaceful face to an explosive or anxious one, representing the arousal dimension. The third row ranges from a very small insignificant figure to a ubiquitous, pervasive one, representing the dominance affective dimension.

We conducted the analysis using R 3.6.0 [54].

## 3. Results

Table 1 summarizes the final version of the MoVIES (the full version is available on request). As shown, 27 items relate to the form of the suicide scene under examination, and 14 items assess the compliance of the scene’s content with the WHO recommendations.

### 3.1. Sample of Suicide Scenes

Regarding the type of movies, 68% of the suicide scenes were extracted from dramas, 16% from action movies, and 16% from other genres. The audience of each movie ranged from 5.330 to 168 million entries. The characters employed various means of suicide, including high-velocity impact (29%), a firearm (22%), poisoning (17%), and hanging (8%).

### 3.2. Reliability

#### 3.2.1. Inter-Expert Reliability

Table 1 provides Cohen’s kappa for each item of the grid. Cohen’s kappas were greater than 0.60 for most items, and inter-rater agreement ranged from substantial (39.1% of items) to almost perfect (48.8% of items). Only 5 of the 41 (12.2%) items demonstrated moderate to fair inter-rater agreement: Correspondence system (object, decor, accessory with symbolic value in the film), κ = 0.6; scene played with particular emotional intensity by the character (anger, romanticism, sadness exaggerated), κ = 0.3; mention of a mental disorder, κ = 0.3; suicide is presented as a solution, κ = 0.4; and the character explicitly evokes suicidal ideation before acting out, κ = 0.5.

#### 3.2.2. Internal Consistency

Internal consistency, indicated by Cronbach’s coefficient, was calculated based on the sum of the scores obtained in the second series. Cronbach’s alpha was 0.85 [0.31–2.42] for our sample, indicating good consistency of the analysis grid.

### 3.3. Predictive Validity

In the multiple linear regression (see Table 2), identification was significantly associated with the MoVIES grid, valence, and emotional arousal (coefficients were 0.48, 3.19, and −2.65, respectively; *p* < 0.05), but not with empathy (0.19, *p* > 0.05). The average score for the MoVIES grid for the validation sample was 17.56 (on 41 items).

## 4. Discussion

To our knowledge, the MoVIES is the first analytical tool developed to quantitatively analyze video scenes depicting suicide or suicide attempts in terms of their possible effect on the viewer. Based on objective characteristics, it summarizes in a single measure several of the characteristics that the literature has suggested to be involved in the suicidal contagion process. More specifically, it estimates the video’s intrinsic risk of identification with the character who attempts or dies by suicide.

The validation procedure of the MoVIES demonstrated satisfying inter-examiner reliability for almost all items and good internal consistency. Importantly, the MoVIES score significantly predicted the strength of identification with the depicted characters, independent of the viewer’s baseline empathy and emotional arousal/valence after exposure to the video scene. Suggesting that the grid items are actual inductors of the psychological conditions assumed to be the root of suicidal contagion, these findings reinforce the validity of the scale.

The MoVIES can be considered an analogue of the two analytical grids published to characterize how the print media covers suicide stories. The Risk of Imitative Suicide Scale (RISc) [35] and PRINTQUAL both aim to quantify the quality of press articles on suicide facts in terms of compliance to official recommendations for Werther Effect prevention [36]. The MoVIES has similar inter-rater reliability as PRINTQUAL. Common to the three instruments is that they estimate the risk of suicide contagion that can be assigned to the triggering medium. However, in contrast to the two other scales, the MoVIES does not rely on a normative approach. In the fictional cinematographic domain, there is no recommendation or consensus about how to represent suicide to minimize the risk of contagion. Because measures of compliance were not possible, we opted for a theory-informed approach. Based on a comprehensive and trans-disciplinary review of the literature, the MoVIES appraises the risk of contagion through a complex, holistic, causal construct that integrates empathy and social learning components. The notions of “immersion” [55], “identity work,” or “ego-involvement” [56] that cinema characteristics in the MoVIES assessment are expected to elicit meet this embodied simulation allegedly involved in suicidal contagion. In the second section of the grid, the WHO recommendations are used to probe the potential of video scenes in inducing psycho-cognitive components of the suicidal contagion process, most referring to the social learning framework. For instance, depicting the suicide method is assumed to increase the cognitive availability of suicide among vulnerable audiences [57]. Similarly, several items refer to misconceptions that fail to protect suicidal viewers from imitating the character (e.g., “suicide has a unique cause”, “suicide is unpredictable”, or “suicide is not preventable”) [58]. Overall, the good internal consistency of the MoVIES and its predictive validity confirm the coherent integration of the different concepts, which are in agreement with evidence from psychology [46], semiotics [56], sociology [59], and cinema [60].

Several limitations should be considered before using the MoVIES. First, all the items had the same weight in the final score computation, although some may be of specific relevance depending on the video under consideration. Furthermore, we did not conduct a component analysis to probe the dimensionality of the scale. Because it aggregates several theoretical backgrounds, the main challenge of the MoVIES was to capture a single coherent construct that could be objectively measured. Inevitably, this was achieved at the partial expense of qualitative finesse of analysis. However, to answer specific research needs, each item of the MoVIES can still be interpreted separately as a complement of the global score computation. Second, the predictive validity of the grid was tested using a core-hypothesized causal mechanism (identification), rather than a distal outcome (suicidal imitation or behavior) as a dependent variable. In addition, the study was conducted on a non-clinical population for obvious ethical reasons. Rather than the interaction between viewers and films or TV shows, the MoVIES intends to qualify the video content per se. Its ability to predict the identification propensity of the viewers was tested for pure validation purposes. Further “audience reception” research needs to be conducted before considering using the MoVIES’ score as an indicator for the actual risk of suicidal contagion. For instance, it has been suggested that suicidality crucially alters identification tendencies. The character’s resemblance and/or moral conformity with the viewer’s values is also of importance (there is robust evidence that suicide in villains is less likely to generate identification). Finally, one should bear in mind that identification is not the only mechanism involved in the contagion process. If the MoVIES implicitly covers other mechansims, such as cognitive availability, mood alteration, or social learning through the operationalization of the WHO recommendations, the statistic relationship of the MoVIES’ score with cognitive, affective, or behavioral responses remains to be explored. Finally, the MoVIES was merely designed to study at-risk components of most harmful scenes. Papageno-related items are treated as negative correlates of Werther-related items through reverse rating. Consequently, one should note that the MoVIES doesn’t apply to potentially preventive contents, such as videos depicting suicide ideations without self-harm.

## 5. Conclusions

The MoVIES opens tracks for future research on contagion related to videos depicting fictional suicide. By informing epidemiological and audience reception studies, it could help in disentangling the relative contributions of the inherent properties of a video stimulus on the one hand, and the vulnerability of the viewer on the other in the causal processes leading to contagion. Importantly, future research using the MoVIES should be conducted keeping in mind important ethical issues. Unlike journalists for whom scholars have proposed guidelines when reporting suicide, the relevance of editing recommendations for producers, directors, and actors is questionable. The creative process of films or a series is not similar to the objectivity that guides proper information. On the audience side, it should also be clear that instruments such as the MoVIES are not intended to supplant, but rather to promote consultation with suicide prevention experts.

## Figures and Tables

**Figure 1 ijerph-16-02780-f001:**
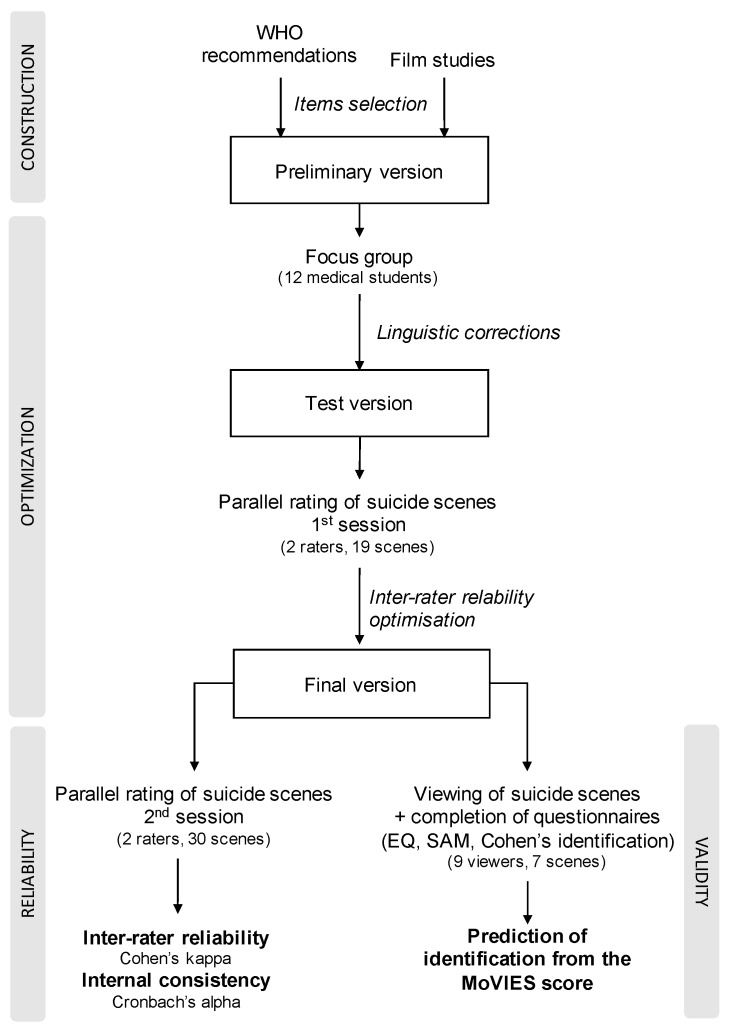
Construction and validation process of the *Movies and Video: Identification and Emotions in Reaction to Suicide* (MoVIES). EQ: Empathy Questionnaire; SAM: Self-Assessment Manikin.

**Table 1 ijerph-16-02780-t001:** The MoVIES items with corresponding inter-rater reliability.

No.	Item Label	κ
	**Rating of the cinematographic form**	
*1*	The suicide scene concerns the main character	0.92
*2*	The character represents a figure of attachment, a model	0.71
3	Suicide is represented during more than 30% of the scene	0.84
4	Suicide scene duration is longer than one minute	0.86
5	The suicide is represented in a rapid succession of shots	1
6	Multiple shots of the suicide scene are interspersed with shots of another scene	1
7	The suicide scene is introduced with a smash-cut, suggesting suddenness and unexpectedness	1
8	Suicide scene shows an explicit representation of the suicide or dead body	1
9	Suicide scene shows the presence of blood or visible internal organs	1
10	Suicide scene shows a close-up of the corpse	0.95
11	Suicide scene shows a close-up of the means of suicide	0.92
12	Presence of a circular traveling or camera range	0.96
13	Presence of a low angle view	1
14	Presence of a general shot	0.86
15	Presence of slow motion	1
16	Unbalanced composition of the scene	1
17	Presence of a correspondence system (object, decor, accessory with symbolic value in the film)	**0.60**
18	Presence of a sequence “above-the-shoulder”	0.80
19	Offset plane with laterally inclined frame	0.91
20	Presence of camera movement (moving from a medium or overall shot to a close-up)	1
21	Suicide scene includes lighting in chiaroscuro	0.74
22	Suicide scene shows lighting with red, purple, blue color filter	0.85
23	Good realism of the scene	0.73
24	Suicide scene played with particular emotional intensity by the character (anger, romanticism, sadness exaggerated)	**0.37**
25	Presence of dialogs or speeches (words)	0.89
26	Presence of music	1
27	Presence of non-verbal manifestations of anxiety	0.7
	**Rating of the content**	
28	The suicide is presented in a trivial way	0.91
29	The suicide is associated with a single cause	0.92
30	The act could have been predicted by the characters *	0.94
31	Something could have been done to prevent the suicide *	0.76
32	Mention of a mental disorder *	**0.38**
33	Elements of the speech sensationalize, normalize, or criminalize the suicide	0.74
34	The suicide is presented as a solution	**0.49**
35	Detailed information about the method used for the suicide is shown	0.82
36	Type of death (soft or slow death)	0.84
37	People in mourning are represented *	0.95
38	Information is provided about resources from which to get help *	0.91
39	Interventions that could have contributed to preventing the suicide are depicted *	0.80
40	Information about the risk factors or warning signs of suicide is provided *	0.78
41	The character explicitly evokes suicidal ideation before acting out *	**0.55**

* Items related to the Papageno effect, for which a true statement is rated as 0. Κ: Cohen’s kappa. In bold, Cohen’s kappas greater than 0.60.

**Table 2 ijerph-16-02780-t002:** Results of the multivariate analysis to test the association between identification and the MoVIES score, empathy, emotional valence, and emotional arousal.

	Mean (SD)	β	IC 95 %	*p*
MoVIES score	17.56 (5.14)	0.48	[0.04–0.92]	0.032
Baseline empathetic capacity (Empathy questionnaire score)	2.43 (1.20)	0.19	[−0.09 to 0.48]	0.190
Post-exposition Emotional Valence (SAM—valence score)	2.10 (1.62)	3.19	[0.54–5.85]	0.019
Post-exposition Emotional Arousal (SAM—arousal score)	−2.31 (1.38)	−2.65	[−4.812 to −0.49]	0.017

MoVIES: Movies and Video: Identification and Emotions in reaction to Suicide; SAM: Self-Assessment Manikin.

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
