# Peer review of "Construction and Validation of an Analytical Grid about Video Representations of Suicide (“MoVIES”)"

_ijerph, 2019, doi:10.3390/ijerph16152780_

Round 1

Reviewer 1 Report

Thank you for the opportunity to review this interesting paper.  The authors have devised a scale to measure the degree to which people may be likely to identify with characters who are depicted attempting and/or dying by suicide in films.  I think the work has a lot of merit and should be published, however I have substantial concerns about the way in which the results are framed and the strength with which the authors claim that tool is useful.

The authors claim that their instrument can “quantify the risk of suicidal contagion” despite the fact that they have not proven that.  They are quantifying the risk of identification in non-suicidal people (i.e. not the people we are most interested in).  The fact that the test audience were not suicidal at the time, might affect the degree of identification.  Indeed, while I agree that identification in those who are suicidal might have functioned as “a strong acceptable proxy”, I do not think it follows that this is so in the non-suicidal group. 

Furthermore, identification is an important (but not the only) factor which may drive contagion effects.  For example, I was surprised that the model has very little about suicide in villains.  Only the first two questions deal with the kind of character and there is robust research evidence that suicide in villains or other people with whom **viewers are intrinsically unlikely to identify don’t seem to have the same contagion effects.  I could very easily imagine (and think I even saw this weekend) a film which had a villain suicide which would score poorly on this measure but would be unlikely to cause harm. Was this item tested on villains or were all portrayals of protagonist characters?

Moreover, certain items on the scoring system may also be much more important than others.  E.G. I think 13 Reasons Why would have been very different if only two items had changed – that the character had a mental illness and that her death was preventable.  That might have made a huge difference but would have only had a small impact on scoring.  In other words, I think one could argue that if suicide is not presented as preventable, that 1 point may indicate harm almost irrespective of anything else on the scale.

Also, the scale only deals with attempts or deaths.  The literature generally shows that by far the most helpful portrayals of suicide (the ones the Papageno Effect is named after) involve people contemplating suicide and then NOT acting on the thoughts at all except to seek help.  Therefore, this is really a scale to quantify the degree of identification within portrayals that are already the most likely to cause harm (not all portrayals or the most helpful ones).

This is a very serious issue – because we are working hard with the entertainment industry to ask them to have more suicidal characters not act on their thoughts (i.e. Papageno characters) and their focus has really been on finding ways of quickly proving that their work is safe.  They could potentially devise a very harmful suicide scene that nevertheless achieves a good enough score on this measure and then claim to have done a good job.  The bottom line is that these authors have devised an interesting scale which may reliably measure the degree of identification of non-suicidal individuals with the most potentially harmful suicide content in non-villain fictional characters.  That deserves more study, but it is wrong the claim that it can predict contagion and we need to be clear that this should not intended to function as a substitute for consultation with suicide prevention experts – a point which should be made in the abstract, introduction, discussion and conclusions.

More minor points:

While I have expertise in suicide contagion, I am not an expert in scale validation and am uncertain whether 9 subjects are sufficient to claim a scale is valid/leave that to the journal to assess.

Page 2, line 52 “the serious—albeit potentially ambivalent [22-24]—impact of the series”.  I don’t think ambivalent makes sense/is the correct word.  What about “albeit potentially mixed”

Author Response

Thank you for the opportunity to review this interesting paper.  The authors have devised a scale to measure the degree to which people may be likely to identify with characters who are depicted attempting and/or dying by suicide in films.  I think the work has a lot of merit and should be published, however I have substantial concerns about the way in which the results are framed and the strength with which the authors claim that tool is useful.

We deeply thank Reviewer 1 for the value and relevance of his/her comments. We acknowledge the opportunity he/she gave us to improve the paper.

The authors claim that their instrument can “quantify the risk of suicidal contagion” despite the fact that they have not proven that.  They are quantifying the risk of identification in non-suicidal people (i.e. not the people we are most interested in). The fact that the test audience were not suicidal at the time, might affect the degree of identification.  Indeed, while I agree that identification in those who are suicidal might have functioned as “a strong acceptable proxy”, I do not think it follows that this is so in the non-suicidal group. Furthermore, identification is an important (but not the only) factor which may drive contagion effects.  For example, I was surprised that the model has very little about suicide in villains.  Only the first two questions deal with the kind of character and there is robust research evidence that suicide in villains or other people with whom **viewers are intrinsically unlikely to identify don’t seem to have the same contagion effects.  I could very easily imagine (and think I even saw this weekend) a film which had a villain suicide which would score poorly on this measure but would be unlikely to cause harm. Was this item tested on villains or were all portrayals of protagonist characters?

We thank Reviewer 1 for this important remark.

We agree that claiming that the MoVIES measures the risk of suicidal contagion is not only an overstatement, but also a epistemological fallacy. For now, the MoVIES belongs to the “film production” rather than to the “audience reception” realm. Differently said, the MoVIES’ score characterizes nothing but the video productions per se. Although we tested the predictive value of the scale on the viewer’s identification propensity for validation purposes, evidence is still too poor to support its ability in predicting such a complex behavior as suicidal imitation is. This is why we toned down our statements related to suicidal contagion in the abstract [lines 17-18], objective [lines 85-91], discussion [lines 234-236] and conclusion [lines 296-297]. In addition, we added a significant development in the limitation sections to explain how the MoVIES could be further tested to actually serve a predictor for suicidal contagion.

We acknowledge that baseline suicidality is probably one of the strongest audience-related determinant for contagion, most likely by increasing the strength of identification to the suicidal character. One can hypothesize that if the MoVIES predicts identification in a healthy sample, the effect size would be even stronger in a distressed sample. However, we recognize that this remains to be tested and stated it in the manuscript [lines 277-282].

According to us, the question of identification to villains is also more a matter of “audience reception” than of “video content”. Indeed, we agree that several authors (eg. Stack, Kral and Borowski, 2014) has proposed that evil characters systematically generate poorer identification as compared to heroes. However, evidence are debatable and the literature brings nuances. Indeed, "antiheroes" with morally questionable habits are increasingly present in TV shows, and some studies suggest they might paradoxically arise strong identification (Mittell, 2015). Consequently, it is possible that the “interaction” (in terms of value matching, resemblance or attachment) between the character and the viewer is of greater relevance than the nature of the character per se. We added lines about this specific question in the discussion [lines 283-285].

We hope that Reviewer 1 will note that if the MoVIES doesn’t allow for answering all these audience-reception questions from the current study (which we made clear in the manuscript [lines 50, 86, 280-283, 297, 303), he/she will also recognize that the scale is an important and unprecedented step in this direction.

Moreover, certain items on the scoring system may also be much more important than others.  E.G. I think 13 Reasons Why would have been very different if only two items had changed – that the character had a mental illness and that her death was preventable.  That might have made a huge difference but would have only had a small impact on scoring.  In other words, I think one could argue that if suicide is not presented as preventable, that 1 point may indicate harm almost irrespective of anything else on the scale.

This interesting remark points out the balance we had to find between building a coherent, objectifiable construct from the aggregation of heterogeneous elements and the preservation of some important qualitative finesses. Weighting the items is indeed an option we have considered, as well as this of separating Werther-related and Papageno-related items (such as the ones you cite for 13RW). However, this would likely have corrupted the MoVIES’ global score with subjectivity, altered the inter-rater reliability, and decreased the internal consistency in a way that would have been hard to justify (as empirical evidence about the psycho-social mechanisms underlying the suicidal contagion process are still lacking). However, the MoVIES still offers possibilities for finer qualitative inquiries, since each item can still be studied separately in complement of the global score analysis. We discussed this point in the limitation section of the manuscript [lines 273-275].

Also, the scale only deals with attempts or deaths. The literature generally shows that by far the most helpful portrayals of suicide (the ones the Papageno Effect is named after) involve people contemplating suicide and then NOT acting on the thoughts at all except to seek help.  Therefore, this is really a scale to quantify the degree of identification within portrayals that are already the most likely to cause harm (not all portrayals or the most helpful ones).This is a very serious issue – because we are working hard with the entertainment industry to ask them to have more suicidal characters not act on their thoughts (i.e. Papageno characters) and their focus has really been on finding ways of quickly proving that their work is safe.  They could potentially devise a very harmful suicide scene that nevertheless achieves a good enough score on this measure and then claim to have done a good job. 

We thank Reviewer 1 for having raised this point. For sake of internal consistency, we had to treat the Papageno effect as a negative correlate of the Werther effect, which we recognize to be a debatable approximation. In the manuscript, we changed the formulations related to the type of video scenes to which the MoVIES is intended to apply to make it clear that only suicide attempts and suicide deaths are concerned. In addition, we specified in the discussion that the MoVIES does not apply to potentially preventive scene or components [lines 233-236, 275-277, 291-293].

The bottom line is that these authors have devised an interesting scale which may reliably measure the degree of identification of non-suicidal individuals with the most potentially harmful suicide content in non-villain fictional characters.  That deserves more study, but it is wrong the claim that it can predict contagion and we need to be clear that this should not intended to function as a substitute for consultation with suicide prevention experts – a point which should be made in the abstract, introduction, discussion and conclusions.

This point is absolutely not questionable. We outlined it in the conclusion [lines 17-18, 90-91, 100, 235-236, 296, 304-305].

While I have expertise in suicide contagion, I am not an expert in scale validation and am uncertain whether 9 subjects are sufficient to claim a scale is valid/leave that to the journal to assess.

The number of 9 subjects applies to the regression model that allows to study the link between the Identification score and the Movies score. Each subject having viewed 7 scenes, we obtain 63 observations, which is enough information to compute the model.

Page 2, line 52 “the serious—albeit potentially ambivalent [22-24]—impact of the series”.  I don’t think ambivalent makes sense/is the correct word.  What about “albeit potentially mixed”

Thank you for the correction. We replaced “ambivalent” by “mixed” line 52.

Bibliography

Mittell, J. (2015). Lengthy interactions with hideous men: Walter White and the serial poetics of television anti-heroes. In Storytelling in the media convergence age (pp. 74-92). Palgrave Macmillan, London.

Stack, S., Kral, M., & Borowski, T. (2014). Exposure to suicide movies and suicide attempts: A research note. Sociological Focus47(1), 61-70.

Reviewer 2 Report

Authors failed to include limitations.  Authors only had 2 raters for inter-rater reliability which is too low a number for me to say that their new model will be generalizable to the wider public.

Author Response

Authors failed to include limitations. 

We thank Reviewer 2 for his/her comments.

As suggested, we reinforced the limitation section, mainly in response to Reviewer 1’s wonders. More specifically, we brought precision to the scope of application of the MoVIES and acknowledged some metrological limits that results from the validation methods we used [lines 268-293].

Authors only had 2 raters for inter-rater reliability which is too low a number for me to say that their new model will be generalizable to the wider public.

The inter-rater reliability is a measure of how much a scale is reliable, i.e. objectively usable. If a  grid is poorly constructed, raters will put different answers for the same object. On the contrary, the good levels of reliability we found for the MoVIES’ items indicate that the scale doesn’t leave room for subjectivity, so as the investigators gave similar answers for the same scenes.

Inter-rater reliability is not related to generalizability matters. It is usually calculated from 2 observers. Although statistical methods exist, rating procedures with more than 2 judges are rare, and, in any case, do not improve the generalizability of the scale.